# Ultrasonographic findings in patients with abdominal symptoms or trauma presenting to an emergency room in rural Tanzania

Max Bauer[1,2], Faraja Kitila[3], Ipyana Mwasongwe[3], Issa S. Abdallah[3], Evelyne Siongo[3], Sylvester Kasunga[3], Winfrid Gingo[4], Robert Ndege[5], Karin Hasler[6], Daniel H. Paris[1,2], Maja Weisser[1,2,5,7], Martin Rohacek[1,2,3,5]*

1 Department of Medicine, Swiss Tropical and Public Health Institute, Allschwil, Switzerland, 2 University of Basel, Basel, Switzerland, 3 Emergency Department, Saint Francis Referral Hospital, Ifakara, United Republic of Tanzania, 4 Directorate and Surgical Department, Saint Francis Referral Hospital, Ifakara, United Republic of Tanzania, 5 Ifakara Health Institute, Ifakara, United Republic of Tanzania, 6 Emergency Department, Spital Affoltern, Affoltern, Switzerland, 7 Division of Infectious Diseases and Hospital Epidemiology, University Hospital Basel, Basel, Switzerland

* mrohacek@ihi.or.tz

**Data Availability Statement:** All relevant data are within the manuscript.

## Abstract

### Background

Frequencies of ultrasonographic findings and diagnoses in emergency departments in sub-Saharan Africa are unknown. This study aimed to describe the frequencies of different sonographic findings and diagnoses found in patients with abdominal symptoms or trauma presenting to a rural referral hospital in Tanzania.

### Methods

In this prospective observational study, we consecutively enrolled patients with abdominal symptoms or trauma triaged to the emergency room of the Saint Francis Referral Hospital, Ifakara. Patients with abdominal symptoms received an abdominal ultrasound. Patients with an abdominal or thoracic trauma received an Extended Focused Assessment with Ultrasound in Trauma (eFAST).

### Results

From July 1st 2020 to June 30th 2021, a total of 88838 patients attended the emergency department, of which 7590 patients were triaged as 'very urgent' and were seen at the emergency room. A total of 1130 patients with abdominal symptoms received an ultrasound. The most frequent findings were abnormalities of the uterus or adnexa in 409/754 females (54.2%) and abdominal free fluid in 368 (32.6%) patients; no abnormality was found in 150 (13.5%) patients. A tumour in the abdomen or pelvis was found in 183 (16.2%) patients, an intrauterine pregnancy in 129/754 (17.1%) females, complete or incomplete abortion in 96 (12.7%), and a ruptured ectopic pregnancy in 32 (4.2%) females. In males, most common diagnosis was intestinal obstruction in 54/376 (14.4%), and splenomegaly in 42 (11.2%). Of 1556 trauma patients, 283 (18.1%) received an eFAST, and 53 (18.7%) had positive

**Funding:** The authors received no specific funding for this work.

**Competing interests:** The authors have declared that no competing interest exists.

findings. A total of 27 (9.4%) trauma patients and 51 (4.5%) non-trauma patients were sent directly to the operating theatre.

## Conclusion

In this study, ultrasound examination revealed abnormal findings for the majority of patients with non-traumatic abdominal symptoms. Building up capacity to provide diagnostic ultrasound is a promising strategy to improve emergency services, especially in a setting where diagnostic modalities are limited.

## Introduction

Emergencies and acute conditions largely contribute to the mortality and disability of patients in low- and middle-income countries (LMICs) [1]. Worldwide, about 90% of all deaths due to injury occur in LMICs [2]. The implementation of emergency services decreased mortality in sub-Saharan Africa [3–6]. However, physical access to emergency hospital care provided by the public sector in Africa remains poor and varies substantially within and between countries [7]. Therefore, systematic establishment of emergency medical services is urgently needed [8].

In a previous study done at the Saint Francis Referral Hospital (SFRH) in Ifakara, rural South-western Tanzania, 10% of the patients attending the emergency department (ED) suffered from trauma and 6% from abdominal disorders [9]. Point-of-care ultrasound (POCUS) has proven to be a reliable, fast, non-invasive, and cost-effective tool to support diagnosis and improve management of patients suffering from trauma and acute abdominal symptoms [10–16]. Particularly in settings where access to other radiologic modalities is limited due to financial restrictions and challenges in maintenance, POCUS is often the only feasible imaging modality and therefore a promising strategy to improve emergency services [17–25].

The frequencies of ultrasonographic findings and diagnoses in patients with abdominal symptoms or trauma in EDs in sub-Saharan Africa is unknown. This information is important for emergency physicians to determine pre-test probabilities before performing diagnostic ultrasound. Additionally, this knowledge helps to assess the impact of ultrasound examinations on patient management in the ED.

This study aimed to describe the frequency of different findings detected by ultrasound in patients with abdominal symptoms or with trauma presenting to the emergency room (ER) of a referral Hospital in rural Tanzania.

## Materials and methods

### Study design and setting

This prospective observational single centre study was performed at the ED of the SFRH, which serves as a referral centre for a rural population of more than one million people living in the Kilombero valley, in the Southwest of Tanzania [9]. The SFRH has 360 beds and specialized services in internal medicine, heart- and lung diseases, pediatrics, surgery, obstetrics, neonatology and gynaecology, ophthalmology, HIV and tuberculosis care, and emergency services.

Patients visiting the ED are triaged according to the South African Triage Score (SATS) [26]. Patients classified as very urgent are triaged to the ER and managed immediately. All other patients wait to be seen by one of three ED doctors. The ER can manage up to eight

patients at once and has a three-shift system for 24 hours a day. It is equipped with five monitors comprising an electrocardiogram, an oximeter, and a system for non-invasive measurement of blood pressure. In addition, the ER features equipment to perform point-of-care tests (i.e., malaria rapid test, urine pregnancy test, HIV rapid test, and a Point-of-Care i-STAT 1 system (Abbott, Chicago, USA)) as well as an ultrasound machine (Chison EBit 60) [27]. The ultrasound machine is permanently available for attending clinicians and can be moved around. Patients' information is recorded in a registry stored at the ER (name, sex, age, diagnosis, management) and in the electronic patient data system of the hospital.

Five clinicians working permanently at the emergency department were trained in emergency POCUS during courses and everyday practice by certified instructors. After passing an exam, these clinicians were board-certified by the European Federation of Societies for Ultrasound in Medicine and Biology (EFSUMB). Two other doctors had advanced skills in performing POCUS. The four to eight intern doctors rotating to the emergency department received basic training in POCUS by the attending clinicians. In addition to their supervisory function, two certified instructors in abdominal ultrasound and POCUS also performed ultrasound themselves.

The study was conducted in accordance with the Declaration of Helsinki. Both the ethics committees of the Ifakara Health Institute (Institutional Review Board, IHI/IRB/No 08–2020) as well as the National Institute for Medical Research, Tanzania (Ref. NIMR/HQ/R.8aVol.IX/3406), approved the study. Both ethical committees waived written informed consent. Patients were asked to give oral consent to be included in this study.

## Study population

Adults and children with abdominal symptoms (i.e. abdominal pain, diarrhoea, vomiting, hematemesis, melena, vaginal bleeding, abdominal distension, or pregnancy-related problems) or with a blunt or penetrating abdominal or thoracic trauma presenting to the ER from July 1st, 2020, to June 30th, 2021, were eligible and consecutively enrolled. Patients who refused to participate were excluded from the study.

## Study procedures

Sonography was done using a Chison EBit60 ultrasound machine equipped with an abdominal probe (6.0 MHz micro-convex probe C3-E) and a linear probe (5.0–12.0 MHz linear probe L12-E). Ultrasound examinations were performed 24 hours a day, including weekends. Patients without health insurance were charged a fee of 7 US$ for the examination; patients unable to pay were examined for free. Sonographies were conducted by the attending clinician of the ER. If they were performed by an unexperienced clinician (i.e., intern doctor), findings were confirmed by a certified sonographer. If no certified sonographer was on duty (i.e., if an intern doctor was working the night shift), patients with abdominal disorder or trauma admitted to the ward received ultrasound the next morning by a certified sonographer. Ultrasound examinations of patients with an abdominal disorder included imaging of the liver, gallbladder, biliary tract, spleen, pancreas, intestine, large blood vessels including inferior vena cava, kidneys, ureters, urinary bladder, uterus, adnexa, prostate, retroperitoneum, and heart, and pleural- and pericardial space. In pregnant women, confirmation of intrauterine pregnancy, gestational age, foetal cardiac activity, foetal lie, and placenta location were assessed. Since a problem-oriented approach was used, not all organs were always scanned. Patients with suspected blunt or penetrating abdominal or thoracic trauma were examined according the Extended Focused Assessment with Sonography in Trauma (eFAST) protocol [28]. Diagnostic

criteria for pneumothorax was absence of lung sliding, absence of comet tail artefacts, absence of the sea-shore sign in M-mode, or demonstration of lung point [29].

The sonographic findings were used to support or modify the clinical diagnosis and to determine further management such as urgent transfer to the surgical theatre or admission to the ward.

After the examination, baseline data, sonographic findings, and diagnosis were documented in a standardized electronic form using the free online software Epidata 2.1b (The EpiData Association, Odense, Denmark). Abdominal ultrasound and eFAST were reported in two different forms including age, sex, history (i.e., duration of symptoms or trauma mechanism), vital signs, clinical findings, results of point-of care-tests, sonographic findings, and diagnosis. In the forms, baseline characteristics were coded as continuous (i.e., age) or binary (i.e., sex) variables. Clinical and sonographic findings were assessed as binary variables (0 = normal, 1 = abnormal). Findings and diagnosis were further specified in a free text field. After filling out the form, the report was stored electronically and printed on paper to be added to the medical file.

## Statistical analysis

The data was extracted from Epidata to an Excel spreadsheet. The analysis of the continuous and binary variables was done using Microsoft Excel software. Free text fields were analysed by the authors to describe the frequency of different diagnoses. The number, diagnoses and management of all patients attended at the ER were obtained from the separate registry of the ER. The total amount of patients visiting the ED was extracted from the electronic patient data system of the hospital (Electronic Health Management System (eHMS), http://gpitg.co.tz/ehms/).

## Results

From July 1$^{st}$, 2020, to June 30$^{th}$, 2021, a total of 88838 patients attended the ED of the SFRH. Of these, 81248 (91.5%) patients had a low triage early warning score of 0–4 and no serious condition. A total of 7590 (8.5%) were labelled as urgent and were triaged to the emergency room. A total of 5289 (69.7%), were adults and adolescents aged ≥14 years, and 2301 (30.3%) were children aged <14 years. Of patients seen at the ER, 4611 (60.8%) were admitted to the wards or transferred to the surgical theatre, and 2979 (39.2%) were managed as out-patients. Table 1 shows the diagnoses of the 7590 patients managed at the ER: A total of 1605 patients suffered from a non-traumatic abdominal- or gynaecological condition, and 1556 had a trauma related condition.

Sonography was performed in 1523 (20.1%) of these 7590 patients. Out of these, 110 (7.2%) patients were excluded because they did not meet the inclusion criteria (i.e., they presented with non-abdominal symptoms or without trauma). Table 2 shows the characteristics of the 1130 (14.9%) non-trauma patients who received an abdominal ultrasound and of the 283 (18.2%) of 1556 trauma patients who received an eFAST. Patients with abdominal symptoms had a median age of 35 years (range 1–96), were mostly female (67%) and had a main complaint of abdominal pain in 87.9% of all cases, followed by amenorrhea or vaginal bleeding in 41.9% and 26.4% of all females. Trauma patients were young (median age 28 years (range 1–83), predominantly male (79.9%) and presented to the ED with history of motor traffic accident (62.9%), violence (13.8%), fall from a tree (10.6%) or an animal attack (1.4%).

Table 3A and 3B show the findings of patients receiving an abdominal ultrasound or an eFAST. Most common findings in non-trauma patients were abnormalities of the uterus or adnexa in 409 of 754 (54.2%) females, and abdominal free fluid in 253 (33.6%) females and 115

**Table 1. Diagnoses of 7590 patients seen at the emergency room of the Saint Francis referral hospital in Ifakara, Tanzania from July 1st, 2020, to June 30th 2021.**

| | Total N = 7590 | Outpatients ≥14 years N = 2063 | Inpatients ≥14 years N = 3226 | Outpatients <14 years N = 916 | Inpatients <14 years N = 1385 |
|---|---|---|---|---|---|
| Infectious diseases[a], n (%) | 2175 (28.7) | 297 (3.9) | 564 (7.4) | 501 (6.6) | 813 (10.7) |
| Other medical conditions[b], n (%) | 1645 (21.7) | 438 (5.8) | 774 (10.2) | 124 (1.6) | 309 (4.1) |
| Trauma[c], n (%) | 1556 (20.5) | 598 (7.9) | 546 (7.2) | 219 (2.9) | 193 (2.5) |
| Abdominal conditions[d], n (%) | 804 (10.6) | 333 (4.4) | 346 (4.6) | 69 (0.9) | 56 (0.7) |
| Gynaecological conditions[e], n (%) | 801 (10.6) | 176 (2.3) | 625 (8.2) | | |
| Cardiovascular conditions[f], n (%) | 482 (6.4) | 167 (2.2) | 298 (3.9) | 3 (0.04) | 14 (0.2) |
| Psychiatric conditions[g], n (%) | 127 (1.7) | 54 (0.7) | 73 (1) | 0 | 0 |

[a] Malaria, respiratory tract infection, sepsis, undifferentiated febrile illness, tuberculosis

[b] Medical conditions which are not of infectious, cardiac, or abdominal origin such as asthma, seizure, stroke, malnutrition, endocrinological diseases, haematological diseases, kidney injury, chronic obstructive pulmonary disease, allergies

[c] Blunt or penetrating trauma, burn

[d] Gastroenteritis, intestinal obstruction, peptic ulcer disease, liver cirrhosis, urinary tract infection, urinary retention, urolithiasis, appendicitis, hernia, peritonitis, abdominal tumour

[e] Pregnancy related condition, mass, pelvic inflammatory disease

[f] Heart failure, hypertensive emergency

[g] Psychotic episode, panic attack

**Table 2. Characteristics of 1130 patients who received abdominal ultrasound and 283 patients who received eFAST.**

| | Patients with abdominal symptoms N = 1130 | Patients with trauma N = 283 |
|---|---|---|
| **Demographics** | | |
| Age, years, median (range) | 35 (1–96) | 28 (1–83) |
| Aged below 14 years, n (%) | 55 (4.9) | 50 (17.7) |
| Male sex, n (%) | 376 (33) | 226 (79.9) |
| Female sex, n (%) | 754 (67) | 57 (20.1) |
| Referral from another hospital, n (%) | 265 (20) | 75 (26.5) |
| Days from accident, median (range) | | 0.4 (0–7) |
| **Vital signs** | | |
| BP systolic < 90mmHg, n (%) | 39 (3.5) | 10 (3.5) |
| BP systolic mmHg median (range) | 124 (56–266) | 123 (62–218) |
| BP diastolic mmHg median (range) | 76 (20–175) | 76 (40–122) |
| HR > 100 BPM, n (%) | 365 (32.3) | 67 (23.7) |
| HR (BPM), median (range) | 90 (52–199) | 90 (32–170) |
| SpO2 < 90%, n (%) | 48 (4.2) | 10 (3.5) |
| Respiratory Rate > 20/min, n (%) | | 57 (20.1) |
| GCS 15 to 13, n (%) | | 255 (90.1) |
| GCS 12 to 9, n (%) | | 18 (6.4) |
| GCS 8 and below, n (%) | | 10 (3.5) |
| **Complaints** | | |
| Duration of symptoms days, median (range) | 19.7 (0–665) | |
| Fever (>37.7˚C), n (%) | 88 (7.8) | |

(*Continued*)

**Table 2.** (Continued)

| | Patients with abdominal symptoms N = 1130 | Patients with trauma N = 283 |
|---|---|---|
| Abdominal pain, n (%) | 993 (87.9) | 83 (29.3) |
| Diarrhea, n (%) | 58 (5.1) | |
| Constipation, n (%) | 85 (7.5) | |
| Vomiting, n (%) | 240 (21.2) | |
| Vomiting blood, n (%) | 30 (2.7) | |
| Melena, n (%) | 41 (3.6) | |
| Amenorrhea, n (% of women) | 316 (41.9) | |
| Vaginal bleeding, n (% of women) | 199 (26.4) | |
| Vaginal discharge, n (% of women) | 75 (9.9) | |
| Chest pain, n (%) | 98 (8.7) | 92 (32.5) |
| Dyspnea, n (%) | 88 (7.8) | 17 (6.0) |
| Pelvic pain, n (%) | | 52 (18.4) |
| **Admission** | | |
| Admitted to ward n (%) | 712 (63) | 171 (60.4) |
| **Findings** | | |
| Airway not patent, n (%) | | 13 (4.6) |
| Diminished breath sound, n (%) | | 8 (2.8) |
| Crackles, n (%) | | 7 (2.5) |
| **Type of accident** | | |
| Motor traffic accident, n (%) | | 178 (62.9) |
| Violence, n (%) | | 39 (13.8) |
| Fall from tree, n (%) | | 30 (10.6) |
| Animal attack, n (%) | | 4 (1.4) |
| Other mechanism, n (%) | | 32 (11.3) |

BP, blood pressure; HR, heart rate; BPM, beats per minute; GCS, Glasgow coma scale score

of 376 (30.6%) males. A total of 85 (22.6%) males had an intestinal abnormality, and 82 (21.8%) had a liver abnormality. In 150 (13.3%) patients, the ultrasound was normal.

In trauma patients, eFAST was abnormal in 53 patients (18.7%), with abdominal free fluid being the most common positive finding (n = 38, 13.4%). Patients with a positive eFAST were most likely to have a history of a motor traffic accident (n = 25, 47.2%), violence (n = 15, 28.3%), and a fall from a tree (n = 10, 18.9%).

In 1130 non-trauma patients who were examined with abdominal ultrasound, 1326 diagnoses were made (Table 4 and Fig 1): Most common diagnoses were an obstetric-gynaecological disorder, an intestinal disorder, or a tumour. A tumour in the abdomen or pelvis was found in 183 of 1130 (16.2%) patients. An intrauterine pregnancy was found in 129 of 754 (17.1%) females, complete or incomplete abortion in 96 (12.7%) females, and a ruptured ectopic pregnancy in 32 (4.2%) females. In males, most common diagnosis was intestinal obstruction in 54 of 376 (14.4%), and splenomegaly in 42 (11.2%). Of the excluded patients, most had pneumonia (n = 82/110, 74.5%) or heart failure (n = 16/110, 14.5%).

Of the 53 patients with a positive eFAST, a total of 27 (50.9%) were sent directly from the ER to the surgical theatre. A total of 51 patients who received an abdominal ultrasound were transferred directly from the ER to the surgical theatre, most of them with a ruptured ectopic pregnancy (n = 32, 62.7%), an intestinal obstruction (n = 6, 11.8%), or a bowel perforation (n = 6, 11.8%).

**Table 3. A. Frequency of different findings in 1130 abdominal ultrasound examinations. B. Frequency of different findings in 283 eFAST examinations.**

**A**

|  | Total N = 1130 | Females N = 754 | Males N = 376 |
|---|---|---|---|
| Abnormalities of uterus or adnexa[a], n (%) | 409 (36.2) | 409 (54.2) |  |
| Abdominal free fluid, n (%) | 368 (32.6) | 253 (33.6) | 115 (30.6) |
| Intestinal abnormality[b], n (%) | 145 (12.8) | 60 (8) | 85 (22.6) |
| Liver abnormality[c], n (%) | 132 (11.7) | 50 (6.6) | 82 (21.8) |
| Kidney abnormality[d], n (%) | 119 (10.5) | 57 (7.6) | 62 (16.5) |
| Splenic abnormality[e], n (%) | 117 (10.3) | 62 (8.2) | 55 (14.6) |
| Pleural fluid, n (%) | 88 (7.8) | 42 (5.6) | 46 (12.2) |
| Gallbladder/biliary abnormality[f], n (%) | 76 (6.7) | 38 (5) | 38 (10.1) |
| B-lines[g], n (%) | 69 (6.1) | 32 (4.2) | 37 (9.8) |
| Inferior vena cava abnormality[h], n (%) | 68 (6.0) | 37 (4.9) | 31 (8.2) |
| Cardiac abnormality[i], n (%) | 66 (5.8) | 33 (4.4) | 33 (8.8) |
| Abnormal urinary bladder, ureter, prostate [j], n (%) | 59 (5.2) | 18 (2.3) | 41 (10.9) |
| Pericardial fluid, n (%) | 50 (4.4) | 26 (3.5) | 24 (6.4) |
| Hydronephrosis, n (%) | 49 (4.3) | 22 (2.9) | 27 (7.2) |
| Portal vein abnormality[k], n (%) | 47 (4.2) | 21 (2.8) | 26 (6.9) |
| Pancreatic abnormality[l], n (%) | 23 (2) | 10 (1.3) | 13 (3.5) |
| Lymph nodes >1.5cm retroperitoneal | 19 (1.7) | 13 (1.7) | 6 (1.6) |
| No abnormal findings, n (%) | 150 (13.5) | 96 (12.7) | 54 (14.4) |

**B**

|  | Total N = 383 | Females N = 57 | Males N = 226 |
|---|---|---|---|
| Normal, n (%) | 230 (81.3) | 51 (89.5) | 179 (79.2) |
| Abdominal free fluid, n (%) | 38 (13.4) | 5 (8.8) | 33 (14.6) |
| Pleural fluid, n (%) | 17 (6) | 1 (1.8) | 16 (7.1) |
| Abdominal and pleural fluid, n (%) | 6 (2.1) | 1 (1.8) | 5 (2.2) |
| Pneumothorax [m], n (%) | 5 (1.8) | 0 | 5 (2.2) |
| Pericardial fluid, n (%) | 3 (1.1) | 0 | 3 (1.3) |
| Pleural fluid and pneumothorax [m], n (%) | 2 (0.7) | 0 | 2 (0.9) |

eFAST, extended Focused Assessment with Sonography in Trauma

[a] Pregnancy-related findings, mass, cyst

[b] Sings of intestinal obstruction or appendicitis, visceral wall thickening, mass

[c] Hepatomegaly, mass, abnormal echogenicity/structure

[d] Mass, cyst, abnormal echogenicity/structure

[e] Splenic length of >14cm, hypo- or hyperechogenic lesion

[f] Gall stones, gallbladder wall thickening, signs of biliary congestion

[g] Three or more B-lines per field of view

[h] Diameter > 2.0 cm, not collapsing >50% during inspiration

[i] Dilatation or hypertrophy of heart, visually impaired left- or right ventricular systolic function

[j] Mass in bladder or prostate, urolithiasis, dilated ureter

[k] Diameter >13mm, thrombosis, periportal fibrosis

[l] Mass, abnormal echogenicity, pancreatic duct >2mm

[m] Pneumothorax was diagnosed if lung sliding, comet tail artefacts, and seashore sign in M-mode was absent, or if lung point could be demonstrated [29].

Of the included abdominal ultrasounds and eFAST, 1318 (93.3%) were performed by certified sonographers, 40 (2.8%) by an advanced sonographer, and 55 (3.9%) by intern doctors who were supervised by a certified sonographer.

**Table 4. Frequencies of 1326 diagnoses in 1130 patients with an abdominal ultrasound.**

| | All N = 1130 | Female N = 754 | Male N = 376 |
|---|---|---|---|
| **Abdominal disorders, n (%)** | **578 (51.2)** | **211 (28)** | **367 (97.6)** |
| Intestinal obstruction, n (%) | 75 (6.6) | 21 (2.8) | 54 (14.4) |
| Splenomegaly, n (%) | 62 (5.5) | 20 (2.7) | 42 (11.2) |
| Abdominal tumour, n (%) | 43 (3.8) | 18 (2.4) | 25 (6.6) |
| Hepatomegaly, n (%) | 41 (3.6) | 16 (2.1) | 25 (6.6) |
| Peritonitis, n (%) | 40 (3.5) | 18 (2.4) | 22 (5.9) |
| Ascites of unknown origin, n (%) | 40 (3.5) | 13 (1.7) | 27 (7.2) |
| Prostate- or urinary bladder tumour, n (%) | 36 (3.2) | 18 (2.4) | 18 (4.8) |
| Liver cirrhosis without tumour, n (%) | 24 (2.1) | 7 (0.9) | 17 (4.5) |
| Appendicitis, n (%) | 23 (2) | 12 (1.6) | 11 (2.9) |
| Liver mass, n (%) | 22 (1.9) | 11 (1.5) | 11 (2.9) |
| Cholecystolithiasis, choledocholithiasis, n (%) | 22 (1.9) | 9 (1.2) | 13 (3.5) |
| Enteritis or colitis, n (%) | 22 (1.9) | 8 (1.1) | 14 (3.7) |
| Hernia, n (%) | 21 (1.9) | 10 (1.3) | 11 (2.9) |
| Pelvic tumour with hydronephrosis, n (%) | 17 (1.5) | 4 (0.5) | 13 (3.5) |
| Urolithiasis, n (%) | 17 (1.5) | 6 (0.8) | 11 (2.9) |
| Urinary retention, n (%) | 15 (1.3) | 0 | 15 (4) |
| Perforation of intestines, n (%) | 13 (1.2) | 2 (0.3) | 11 (2.9) |
| Nephritis, n (%) | 13 (1.2) | 4 (0.5) | 9 (2.4) |
| Cholecystitis, n (%) | 12 (1.1) | 5 (0.6) | 7 (1.9) |
| Splenic mass, n (%) | 7 (0.6) | 2 (0.3) | 5 (1.3) |
| Liver cirrhosis with liver tumour, n (%) | 5 (0.4) | 3 (0.4) | 8 (2.1) |
| Hepatosplenic schistosomiasis, n (%) | 5 (0.4) | 2 (0.3) | 3 (0.8) |
| Urogenital schistosomiasis, n (%) | 3 (0.3) | 2 (0.3) | 1 (0.3) |
| **Gynecological disorders, n (%)** | **479 (42.4)** | **479 (63.5)** | |
| Abortion (complete/incomplete), n (%) | | 96 (12.7) | |
| Ovarian cyst, n (%) | | 88 (11.7) | |
| Pelvic inflammatory disease, n (%) | | 54 (7.2) | |
| Solid tumor of ovaries, uterus or cervix, n (%) | | 53 (7) | |
| Ruptured ectopic pregnancy, n (%) | | 32 (4.2) | |
| Intrauterine fetal death, n (%) | | 11 (1.5) | |
| Molar pregnancy, n (%) | | 8 (1.1) | |
| Ectopic pregnancy, not ruptured, n (%) | | 5 (0.7) | |
| Pregnancy, 1st trimester, n (%) | | 33 (4.4) | |
| Pregnancy, 2nd trimester, n (%) | | 33 (4.4) | |
| Pregnancy, 3rd trimester, n (%) | | 63 (8.4) | |
| **Heart- or lung disorders, n (%)** | **122 (10.8)** | **69 (9.2)** | **53 (14.1)** |
| Heart failure, n (%) | 74 (6.5) | 43 (5.7) | 31 (8.2) |
| Pneumonia[a], n (%) | 25 (2.2) | 9 (1.2) | 16 (4.3) |
| Lung edema[b], n (%) | 23 (2) | 17 (2.3) | 6 (1.6) |
| **Normal findings on ultrasound, n (%)** | **150 (13.3)** | **80 (10.6)** | **70 (18.6)** |

Multiple diagnoses are possible in one patient

[a] Presence of B-lines in combination with fragmented pleura or subpleural granular artefacts, or infiltrates, and no sign for heart failure

[b] Presence of B-lines in combination with normal pleura

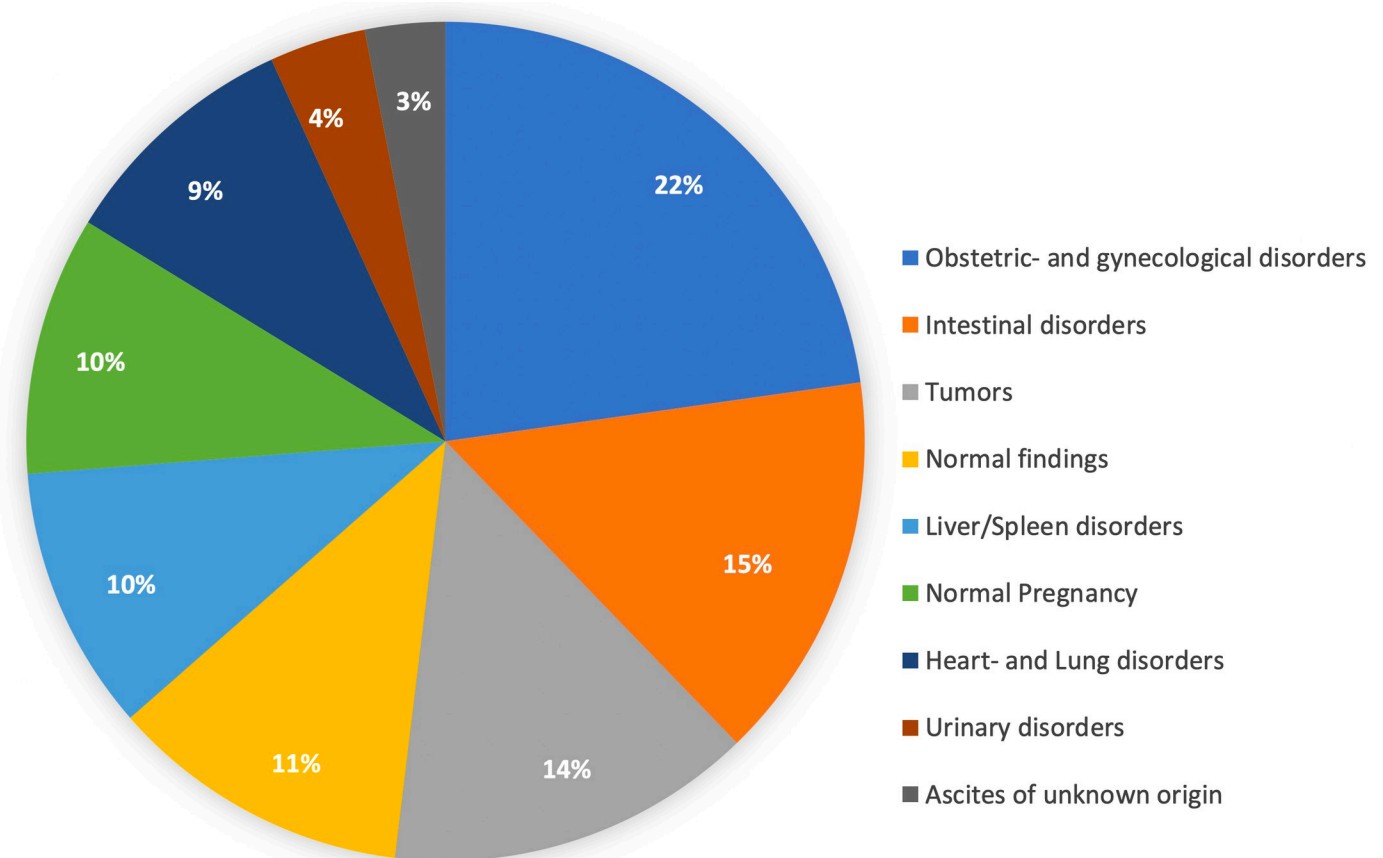

**Fig 1. A total of 1326 diagnoses in 1130 non-trauma patients who received abdominal ultrasound.** The percentages refer to the number of diagnoses.

## Discussion

In this study on ultrasonographic findings and diagnoses in the ER of a rural sub-Saharan African referral hospital, an ultrasound was performed in one fifth of patients presenting to the ER. In patients with a non-traumatic abdominal disorder, only 13% had a normal ultrasound. Abnormalities of the uterus or adnexa and abdominal free fluid were the most common findings. In one of six ultrasound examinations, a tumour of the genitourinary system, liver, spleen, or an abdominal tumour of unknown aetiology was diagnosed. In 32 cases, a ruptured ectopic pregnancy was detected. The most common finding in trauma patients was free abdominal fluid. One in 18 patients who received an ultrasound required urgent surgery.

Other studies evaluating sonography in sub-Sahara Africa showed similar rates of positive findings [30, 31]. However, these studies were performed in radiological departments. The high proportion of patients with an abnormal ultrasound can probably be attributed to presentation with advanced stages of diseases and partly to preselection through referrals from other hospitals. The high frequency of abnormal findings observed in this study confirms the high yield of additional information POCUS can provide in an emergency department in rural sub-Sahara Africa. Most of the patients examined by ultrasound were women. Accordingly, a high incidence of obstetric and gynaecological disorders was found, which has also been shown in other studies from rural Tanzania [32] and is reflected in the high frequency of abnormal sonographic findings of the uterus and ovaries. In other studies assessing sonography in sub-

Sahara Africa, obstetric and gynaecological ultrasound was most frequently performed [31] and most likely changed management [18]. In this study performed in the ER, ultrasound detected important information on pregnancy-related complications such as abortion, ruptured ectopic pregnancy, intra-uterine foetal death, or molar pregnancy. In addition, diagnosis of cystic or solid masses and pelvic inflammatory disease were frequent. Thus, building up sonographic capacity for gynaecological and obstetric disorders should be a high priority also in emergency services.

The burden of cancer in Africa is rising and improved diagnostics for oncological diseases are urgently needed. Abdominal tumours of the liver, cervix, ovary, intestine and urinary tract account for the majority of new cases and contribute largely to fatality [33]. A recent African cohort study on hepatocellular carcinoma (HCC) showed that the majority of patients exhibited a progressed multifocal stage at the time of diagnosis [34]. Since early detection is linked to better survival in cancer patients, improved diagnostic tools and effective surveillance programs are needed to identify patients during early stages of disease [35]. Sonography was shown to have a good sensitivity to detect HCC [36] and is an established diagnostic tool in the staging of gynaecological tumours [37]. In a study from Cameroon, abdominal ultrasound increased the number of diagnoses of Burkitt lymphoma and sonographic staging correlated with the prognosis of patients–hence providing valuable information for further management [38]. In a previous study conducted at the ED of the SFRH, when routine provision of POCUS was not yet established, cancer was diagnosed only in 0.4% of patients [9]. In contrast, this study showed that 16% of patients with abdominal complaints were diagnosed with a potentially malignant tumour. This finding underlines the importance of readily available imaging such as ultrasound to identify oncologic disorders of the abdomen not apparent to clinical examination. In males, most common diagnosis was intestinal obstruction in 14.4%. Although a concomitant tumour was found in one of these patients only, it is possible that more tumours which could not be detected caused intestinal obstruction.

Since other studies either did not differentiate between different abdominal disorders [31] or did not feature a continuous inclusion [17], this is the first study describing the frequency of different abdominal tumours detected by POCUS in an ER in rural sub-Sahara Africa.

Emergency ultrasound has been shown to have a high sensitivity and specificity for diagnosis of intestinal obstruction [39], biliary disease [10], and appendicitis [15]. In our study, sonography identified many acute pathologies of the abdomen leading to diagnoses such as intestinal obstruction, peritonitis, intestinal perforation, or appendicitis. These patients profited greatly from a timely initiation of proper therapy. Thus, albeit we did not perform a comparative study but instead chose a pragmatic descriptive approach in a real-life setting, we believe POCUS is a powerful tool to improve emergency care for patients with acute abdominal symptoms.

Even though this study was primarily aimed at abdominal disorders, pathologies of the chest such as heart failure or pneumonia were also observed in 10% of the included patients. As arterial hypertension is highly prevalent and rarely treated in sub-Saharan Africa [40], many patients present to the emergency department with advanced stages of chronic heart disease [41]. Assessing the inferior vena cava, heart and lungs through sonography allows to quickly differentiate between different causes of dyspnoea [42–44] and can assist in monitoring treatment of heart failure by assessing the severity of pulmonary oedema [45]. Our study was performed during the COVID-19 pandemic, which might have contributed to the frequent diagnosis of pneumonia. In a recent study, lung ultrasound showed a higher sensitivity than chest x-ray for the evaluation of COVID-19 and therefore plays an important part in diagnosis, particularly where access to laboratory services and CT-scans is restricted [46].

In the 2014 WHO report, trauma accounted for 10% of the global burden of disease and for 9% of all deaths [47], of which 90% occurred in LMICs [2]. Addressing emergency care for trauma patients should therefore be a high priority with regard to improving patient outcomes. Due to its high feasibility and practicability and also because of its high specificity in detecting free fluid or pneumothorax, the eFAST protocol was integrated in the Advanced Trauma Life Support (ATLS) guidelines and is frequently used in trauma centres worldwide [48, 49]. In this study, one in five trauma patients undergoing eFAST exhibited abnormal sonographic findings, with free fluid being the most common finding. Half of the patients with positive findings were transferred directly to the surgical theatre from the ER. Studies from trauma centres in South Africa and Nepal found similar rates of positive findings [50, 51]. Using eFAST to examine trauma patients is a valuable tool to identify patients needing urgent surgical care and promises to improve immediate management of patients, especially in settings where CT-scans are unavailable.

To the best of our knowledge, this is the first study describing findings of abdominal ultrasound and eFAST in an ER from a sub-Sahara African hospital with a systematic and consecutive inclusion of patients. As clinicians working at the ED received continuous ultrasound education and were certified by EFSUMB, standardized high-quality sonography was provided day and night and has shown to be feasible.

This study had several limitations: Firstly, due to periods of high patient volume, limited personnel and logistic capacity, some patients did not receive an ultrasound despite meeting the inclusion criteria. Only trauma patients with suspected blunt or penetrating abdominal or thoracic trauma received an eFAST. The relatively low rate of trauma patients receiving sonography can be attributed to a high percentage of burns and isolated limb injuries. Thus, a certain selection bias is probable. Secondly, the study design included neither a follow-up nor a diagnostic comparator such as CT-scan. Therefore, no confirmation of the suspected diagnosis was possible. Thirdly, only a minority of the patients who received abdominal ultrasound were aged under 14 years. Therefore, the utility of emergency ultrasound for paediatric patients could not be assessed exhaustively. Lastly, this was a single centre study and generalizability is limited.

## Conclusion

In rural sub-Sahara Africa, ultrasound was found to frequently detect abnormal findings in severely ill patients presenting to the ER with abdominal symptoms or trauma. Ultrasound helped to make rapid diagnoses of acutely life-threatening diseases such as intestinal obstruction, intestinal perforation, ruptured ectopic pregnancy, pneumothorax, or traumatic abdominal injuries. Moreover, it provided important information for pregnant women and chronic conditions such as intraabdominal and pelvic tumours, liver cirrhosis, or heart failure. Using little resources, POCUS can boost diagnostic capacity and enhance the means of emergency clinicians working in resource-limited settings. Improving both access to and quality of ultrasound is a promising strategy to advance emergency services, especially in a context where other diagnostic modalities are limited.

## Acknowledgments

We thank the emergency department team for supporting the study and the patients for their participation. We thank PD Dr. med. Jan Tuma, Uster, Switzerland, for the training of all who performed ultrasound.

## Author Contributions

**Conceptualization:** Max Bauer, Maja Weisser, Martin Rohacek.

**Formal analysis:** Max Bauer, Martin Rohacek.

**Investigation:** Max Bauer, Faraja Kitila, Ipyana Mwasongwe, Issa S. Abdallah, Evelyne Siongo, Sylvester Kasunga, Winfrid Gingo, Robert Ndege, Karin Hasler, Martin Rohacek.

**Methodology:** Max Bauer, Martin Rohacek.

**Project administration:** Winfrid Gingo, Martin Rohacek.

**Supervision:** Daniel H. Paris, Maja Weisser, Martin Rohacek.

**Writing – original draft:** Max Bauer, Martin Rohacek.

**Writing – review & editing:** Max Bauer, Faraja Kitila, Ipyana Mwasongwe, Sylvester Kasunga, Winfrid Gingo, Robert Ndege, Karin Hasler, Daniel H. Paris, Maja Weisser, Martin Rohacek.

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
