## [Editor Report · Decision Letter 0]

14 Jan 2022

PONE-D-22-00603Diseases detected with sonography in patients with abdominal symptoms or trauma presenting to an emergency room in rural TanzaniaPLOS ONE

Dear Dr. Rohacek,

Thank you for submitting your manuscript to PLOS ONE. After careful consideration, we feel that it has merit but does not fully meet PLOS ONE’s publication criteria as it currently stands. Therefore, we invite you to submit a revised version of the manuscript that addresses the points raised during the review process. Please submit your revised manuscript by Feb 28 2022 11:59PM. If you will need more time than this to complete your revisions, please reply to this message or contact the journal office at plosone@plos.org. Please include the following items when submitting your revised manuscript:A rebuttal letter that responds to each point raised by the academic editor and reviewer(s). You should upload this letter as a separate file labeled 'Response to Reviewers'.A marked-up copy of your manuscript that highlights changes made to the original version. You should upload this as a separate file labeled 'Revised Manuscript with Track Changes'.An unmarked version of your revised paper without tracked changes. You should upload this as a separate file labeled 'Manuscript'.If applicable, we recommend that you deposit your laboratory protocols in protocols.io to enhance the reproducibility of your results. Protocols.io assigns your protocol its own identifier (DOI) so that it can be cited independently in the future. For instructions see: https://journals.plos.org/plosone/s/submission-guidelines#loc-laboratory-protocols. Additionally, PLOS ONE offers an option for publishing peer-reviewed Lab Protocol articles, which describe protocols hosted on protocols.io. Read more information on sharing protocols at https://plos.org/protocols?utm_medium=editorial-email&utm_source=authorletters&utm_campaign=protocols.

We look forward to receiving your revised manuscript.

Kind regards,

Muhammad Abdel-Gawad, MD

Academic Editor

PLOS ONE

Journal Requirements:

Additional Editor Comments:

Title

The word disease is not suitable as it excludes individuals with normal findings. Also, some findings are not unique diseases but a reflection for an existing disease.

Abstract

Please write the aim of the study included in background subsection

Introduction (not mentioned)

Please change the subtitle Background (that is next to abstract) to introduction.

Ethics consideration

All studies conducted on human have to be conducted in accordance with Helsinki standards (please state this in ethics statement)

Table 1

The definition of adults to be less than 14 years is not correct (less than 18 years), please correct

Table 3

How did you diagnose pneumothorax by ultrasound

The footnotes mentioned below table 3 are not related to it, they are belonging to table 2, so please put them below table 2.

Table 4

You mentioned footnotes below it, however, no such notes in the table itself, correct.

In references

For websites, please write the links and accessed date.

Also revise other references to add page range and complete details of cited books, if any.
---

## [Author Response · Author response to Decision Letter 0]

30 Jan 2022

Point by Point Reply

Journal Requirements:

We thank the reviewer for this comment. The manuscript was changed to meet the stated style requirements as shown in the links above. 

- We thank the reviewer for this comment. The ethics statement at the end of the manuscript was deleted. 

- The references were revised. We added Ref 29 on page 7 in the methods section. 

Additional Editor Comments:

Title

The word disease is not suitable as it excludes individuals with normal findings. Also, some findings are not unique diseases but a reflection for an existing disease.

-We thank the reviewer for this valuable comment. “Diseases detected with sonography in patients with abdominal symptoms or trauma presenting to an emergency room in rural Tanzania” 

was changed to “Ultrasonographic findings in patients with abdominal symptoms or trauma presenting to an emergency room in rural Tanzania”

Abstract

Please write the aim of the study included in background subsection Introduction (not mentioned)

- We thank the reviewer for this comment. The aim of the study was included in the Background section of the abstract: “Frequencies of ultrasonographic findings and diagnoses in emergency departments in sub-Saharan Africa are unknown. This study aimed to describe the frequencies of different sonographic findings and diagnosis found in patients with abdominal symptoms or trauma presenting to a rural referral hospital in Tanzania.”

- The sentence “All ultrasound examinations were done by board-certified clinicians in emergency point of care ultrasound.” Was deleted to reduce the wordcount below 300 words.

Please change the subtitle Background (that is next to abstract) to introduction.

- Background was changed to introduction in the text.

Ethics consideration

All studies conducted on human have to be conducted in accordance with Helsinki standards (please state this in ethics statement)

- “The study was conducted in accordance with the Declaration of Helsinki” was included in the ethic statement.

Table 1

The definition of adults to be less than 14 years is not correct (less than 18 years), please correct

- We thank the reviewer for this comment. Since <14 years is the cut off at St. Francis Referral Hospital for being a child, we devided the patients into ≥14 years (adolescents and adults) and <14 years. 

Table 3

How did you diagnose pneumothorax by ultrasound

- We thank the reviewer for this comment. A description and reference defining the diagnosis of pneumothorax was added to the study procedures within the methods section: “Diagnostic criteria for pneumothorax was absence of lung sliding, comet tail artefacts, and sea-shore sign in M-mode, or demonstration of lung point.” On page 7. We added also Ref 29. 

- A description of the diagnosis of pneumothorax was also added to the footnotes of Table 3.

The footnotes mentioned below table 3 are not related to it, they are belonging to table 2, so please put them below table 2.

- -We checked and revised the footnotes of all tables, which are correct now. 

Table 4

You mentioned footnotes below it, however, no such notes in the table itself, correct.

- We apologize for this mistake. We removed tzhe old footnotes, but added new footnotes a and b to define pneumonia and lung edema 

In references

For websites, please write the links and accessed date.

Also revise other references to add page range and complete details of cited books, if any.

- The reference 2 “WHO. Injuries and violence: the facts 2014. World Health Organization Geneva; 2014.» 

was replaced by «World Health Organization. Global Status Report on Road Safety 2009: time for action. Geneva:World Health Organization; 2009. Available from: http://apps.who.int/iris/handle/10665/78256. »

- The reference 3 “Clark M, Spry E, Daoh K, Baion D, Skordis-Worrall J. Reductions in inpatient mortality following interventions to improve emergency hospital care in Freetown, Sierra Leone. 2012.» 

was corrected with the journal and page numbers: «Clark M, Spry E, Daoh K, Baion D, Skordis-Worrall J. Reductions in inpatient mortality following interventions to improve emergency hospital care in Freetown, Sierra Leone. PLoS One. 2012;7(9):e41458.»

- Page numbers were added to the reference 25 «Ngome O, Rohacek M. Point-of-Care Ultrasound: A Useful Diagnostic Tool in Africa. Praxis. 2020;109(8):608-14» 

- Page numbers wree added to the reference 27 «Rohacek M, Hatz C, Weisser M. Emergency Department in the St. Francis Referral Hospital, Ifakara, Tanzania. Praxis. 2017;106(12):651-5.»

- Reference 32 was corrected to «WHO, UNICEF, UNFPA, World Bank Group, UN Population Division. Trends in maternal mortality: 1990 to 2015. Available from: http://apps.who.int/iris/bitstream/10665/194254/1/9789241565141_eng.pdf?ua=1.»

Reference 48 was changed to the book: «American College of Surgeons. 10th Edition of the Advanced Trauma Life Support® (ATLS®) Student Course Manual. Chicago: American College of Surgeons; 2018.»

---

## [Decision Letter · Decision Letter 1]

19 May 2022

Ultrasonographic findings in patients with abdominal symptoms or trauma presenting to an emergency room in rural Tanzania

PONE-D-22-00603R1

Dear Authors,

We’re pleased to inform you that your manuscript has been judged scientifically suitable for publication and will be formally accepted for publication once it meets all outstanding technical requirements.

Kind regards,

Muhammad Abdel-Gawad, MD

Academic Editor

PLOS ONE

Additional Editor Comments (optional):

Reviewers' comments:

Reviewer's Responses to Questions

**Comments to the Author**

1. If the authors have adequately addressed your comments raised in a previous round of review and you feel that this manuscript is now acceptable for publication, you may indicate that here to bypass the “Comments to the Author” section, enter your conflict of interest statement in the “Confidential to Editor” section, and submit your "Accept" recommendation.

Reviewer #1: All comments have been addressed

Reviewer #2: (No Response)

2. Is the manuscript technically sound, and do the data support the conclusions?

Reviewer #1: Yes

Reviewer #2: Yes

3. Has the statistical analysis been performed appropriately and rigorously? 

Reviewer #1: I Don't Know

Reviewer #2: Yes

4. Have the authors made all data underlying the findings in their manuscript fully available?

Reviewer #1: Yes

Reviewer #2: Yes

5. Is the manuscript presented in an intelligible fashion and written in standard English?

Reviewer #1: Yes

Reviewer #2: Yes

6. Review Comments to the Author

Reviewer #1: The manuscript was good and interesting

Introduction was good and included the aim of the study

Materials and methods were good described

Results were good described

Discussion was good written

Reviewer #2: (No Response)

7. PLOS authors have the option to publish the peer review history of their article (what does this mean?). If published, this will include your full peer review and any attached files.

Reviewer #1: No

Reviewer #2: **Yes: **Dr. Sunil Adhikari

---

## [Editor Report · Acceptance letter]

24 May 2022

PONE-D-22-00603R1 

Ultrasonographic findings in patients with abdominal symptoms or trauma presenting to an emergency room in rural Tanzania 

Dear Dr. Rohacek:

I'm pleased to inform you that your manuscript has been deemed suitable for publication in PLOS ONE. Congratulations! Your manuscript is now with our production department. 

Kind regards, 

on behalf of

Prof. Muhammad Abdel-Gawad 

Academic Editor

PLOS ONE